# GeoReasoning: Structured Semantic Reasoning for Image-to-Map Localization

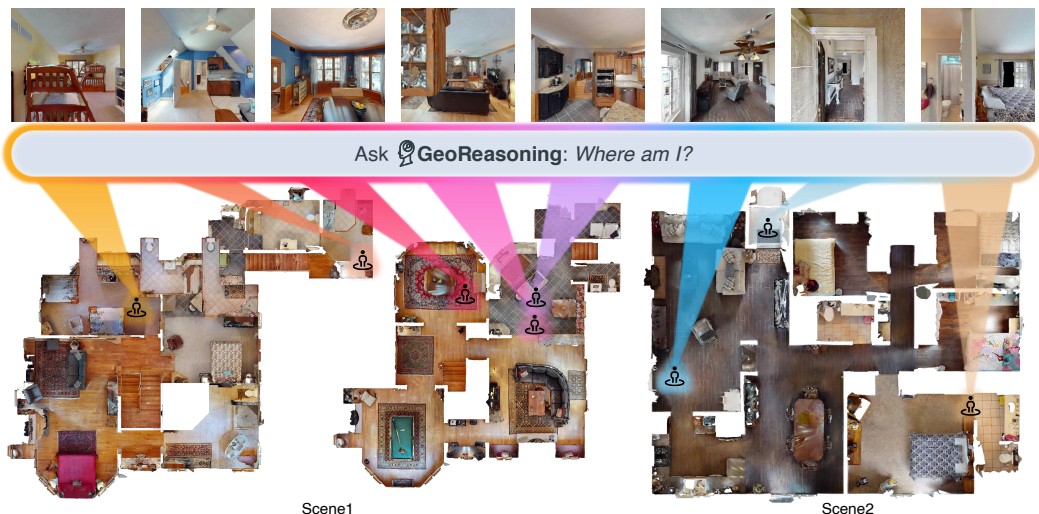

Figure 1: **GeoReasoning** tackles the *reasoning localization* problem by aligning an egocentric RGB view with a 2D floor plan through structured semantic reasoning followed by geometric verification.

## Abstract

We introduce *reasoning localization*, a new paradigm for self-localization that leverages multimodal large language models (MLLMs) to interpret spatial context from 2D maps and first-person images. Unlike traditional approaches that depend on LiDAR, odometry, or engineered markers, reasoning localization emulates how humans orient by aligning visual cues with map structure. To address this new self-localization problem, we present **GeoReasoning**, a zero-shot framework that decomposes image-to-map grounding into *structured semantic reasoning* followed by *geometric verification*. Instead of directly predicting coordinates, GeoReasoning (i) identifies map-visible landmarks, (ii) grounds them as anchors via promptable segmentation, (iii) estimates coarse distances through language-based reasoning, and (iv) solves a robust trilateration program to recover the pose. This design separates high-level semantic reasoning from metric optimization, yielding interpretable rationales, verifiable intermediate outputs, and resilience against map symmetries. To support this task, we release the first benchmark for reasoning localization, spanning diverse indoor maps, image–map pairs, and candidate poses, along with diagnostic metrics such as rationale consistency, mean/median localization error, and success@$r$ for $r \in 0.1, 0.5, 1, 3$ m. Experiments with state-of-the-art MLLMs demonstrate that GeoReasoning significantly improves localization accuracy over direct prediction baselines, while exposing open challenges in symmetry disambiguation and monocular scale estimation. Our results highlight structured reasoning–geometry integration as a promising path toward scalable, human-like localization in GPS-denied settings.

# 1 INTRODUCTION

Localization is a fundamental capability for both humans and robots. In robotics, self-localization typically relies on specialized sensors such as LiDAR, odometry, or artificial markers to navigate and correct drift, which inherently limits scalability and generalization. Humans, by contrast, can localize themselves using only vision and a map. Upon entering an unfamiliar building, we interpret a floorplan or top-down map and align partial visual observations to deduce our position, even in the presence of symmetry, occlusion, or clutter. This natural ability inspires a central question: can embodied AI agents achieve similarly robust, human-level localization using only RGB images and maps, without privileged sensors or environment-specific training?

Directly predicting coordinates from an image and a map is fundamentally ill-posed. The mapping from $(I, M)$ to $(x, y)$ is inherently many-to-one due to map symmetries and monocular scale ambiguities. Free-form textual predictions often ignore pixel-level constraints, relative directions, or spatial relations, producing outputs that are plausible in text but geometrically inconsistent. Unconstrained outputs also lack an explicit optimization objective, making them fragile to noise or incomplete observations. In addition, partial views may reveal only a subset of an object's footprint, and repeated categories in indoor layouts introduce ambiguity. Without explicit verification and relational reasoning, models cannot reliably disambiguate symmetric or visually similar locations.

To address this, we propose *reasoning localization*, a new paradigm that leverages multimodal large language models (MLLMs) for spatial reasoning on 2D maps. Unlike sensor-based approaches, reasoning localization emphasizes *image-to-map grounding* through high-level, test-time reasoning Snell et al. (2024). We introduce **GeoReasoning**, a framework that casts localization as reasoning over semantics to deduce geometry rather than directly predicting coordinates. Given an egocentric RGB observation $I \in \mathbb{R}^{H \times W \times 3}$ and a top-down floor map $M$ (binary or semantic), the goal is to estimate the camera position $p^\star \in \Omega \subset \mathbb{R}^2$ and optionally the yaw angle $\psi$ in a training-free, RGB-only regime. The central idea is to separate what matters, namely landmarks that are stable and map-visible, from how to solve, namely a small, well-posed geometric problem. GeoReasoning follows a two-stage framework that mirrors human behavior: first, the model reasons about salient anchors and grounds them on the map; second, these semantics are converted into metric constraints to solve for the pose. This division of labor allows high-level language reasoning to propose and verify anchors while classical geometry delivers precise and calibrated localization.

GeoReasoning implements this two-stage process in a zero-shot, training-free pipeline. In the first stage, the multimodal LLM identifies map-visible anchors and produces textual descriptions with relational priors encoding object category, distinctive shape, and adjacency. Anchors are grounded on the map using promptable segmentation, yielding masks and centroid points. Indoor layouts often contain repeated categories, and partial views reveal only part of an object, making naive matching brittle. To resolve ambiguities, a cross-view semantic verifier evaluates whether each candidate region corresponds to the same object instance observed in $I$, retaining only consistent anchors. In the second stage, each verified anchor is refined by cropping a local map neighborhood and re-segmenting to improve geometric fidelity. The LLM then estimates camera-to-object distances using monocular cues, producing circle constraints that are integrated via robust trilateration to recover the camera position and optionally orientation. Residuals that cannot be fully minimized produce either a precise pose or a calibrated uncertainty region. Figure 1 shows examples of GeoReasoning, which aligns an egocentric RGB image with a 2D floor plan by reasoning over semantic anchors and refining them through geometric trilateration.

To support systematic evaluation and drive future research, we construct the first benchmark for reasoning localization. It spans diverse indoor environments, egocentric image–map pairs, candidate poses, and diagnostic metrics such as top-$k$ pose accuracy, orientation error, and rationale consistency. Experiments show that structured reasoning with explicit verification significantly improves image-to-map grounding while exposing persistent challenges in symmetry resolution and fine-grained metric alignment.

In summary, our work makes the following contributions:

- We introduce reasoning localization, a new paradigm for self-localization that leverages multimodal large language models to interpret map-level semantics and egocentric observations without requiring privileged sensors or environment-specific training.

- We present GeoReasoning, a zero-shot framework that decomposes localization into structured semantic reasoning and geometric verification, explicitly modeling anchor selection, cross-view consistency, and robust trilateration.

- We release the first benchmark for reasoning localization, including diverse indoor maps, image–map pairs, candidate poses, and diagnostic metrics to evaluate both geometric accuracy and reasoning fidelity.

- We demonstrate that GeoReasoning significantly outperforms direct-prediction baselines while highlighting open challenges such as symmetry disambiguation and monocular scale estimation, emphasizing the value of integrating structured reasoning with geometric optimization.

## 2 RELATED WORK

### 2.1 VISION-BASED LOCALIZATION

Accurate self-localization is a core requirement for mobile robots, traditionally achieved through sensor-driven methods (Zafari et al., 2019). Conventional SLAM approaches fuse data from LiDAR, cameras, and IMUs to build a map and estimate pose (Engel et al., 2014; 2017; Forster et al., 2014; Huang et al., 2024). While effective, these pipelines are sensitive to sensor noise, calibration drift, dynamic scenes, and appearance changes, and long-term deployment requires careful tuning (Cadena et al., 2017).

Recent work has reduced reliance on specialized sensors by leveraging monocular RGB cues enriched with semantics, including objects, rooms, and scene structure, to stabilize data association, loop closure, and global re-localization in indoor spaces (Salas-Moreno et al., 2013; Yang & Scherer, 2019; Nicholson et al., 2018; Taira et al., 2018). Semantic SLAM approaches inject object- or region-level labels to improve robustness and support longer-horizon autonomy. GS4 (Jiang et al., 2025a) demonstrates generalizable sparse-splatting semantic SLAM across multiple datasets without fine-tuning, highlighting the trend toward sensor-light, semantic-aware pipelines. Other methods such as LalaLoc and LalaLoc++ (Howard-Jenkins et al., 2021; Howard-Jenkins & Prisacariu, 2022) learn cross-modal layout embeddings to align panoramic images with 2D floor plans, illustrating the evolution from embedding-based to sparse, generalizable frameworks.

**GeoReasoning** departs from geometry-first estimation by treating localization as reasoning-first map reading. It aligns egocentric RGB views to a floor plan through multimodal language-model reasoning, decomposing localization into explicit reasoning sub-steps and verifying them against global map structure without training or privileged sensors.

### 2.2 SPATIAL REASONING BENCHMARKS FOR MULTIMODAL LLMS

Evaluating spatial reasoning in multimodal LLMs has attracted increasing attention. Early diagnostic datasets (Johnson et al., 2017) tested compositional visual reasoning, revealing shortcuts from dataset biases. Adversarial 2D benchmarks (Yang et al., 2019) assessed spatial relations in real images with balanced positives and negatives, showing that models often exploit superficial cues. Embodied QA benchmarks (Azuma et al., 2022; Ma et al., 2023) evaluate spatial reasoning in indoor 3D scans, while MLLM-focused suites like GSR-BENCH (Rajabi & Kosecka, 2024) and MARBLE (Jiang et al., 2025b) highlight persistent gaps in multistep reasoning under partial observability and complex relational constraints. Most benchmarks, however, focus on relational QA rather than grounding and rarely test egocentric-to-allocentric transformation (Yang et al., 2025; Zhang et al., 2025).

Our benchmark directly addresses this gap. Reasoning localization requires precise alignment of a camera view to a map, assessing both spatial layout understanding and justification of localization decisions. We instantiate it on HM3D (Ramakrishnan et al., 2021), deriving floor plans and egocentric RGB observations from large-scale indoor reconstructions, yielding diverse image–map pairs and candidate poses for systematic zero-shot evaluation, with metrics for both accuracy and rationale consistency.

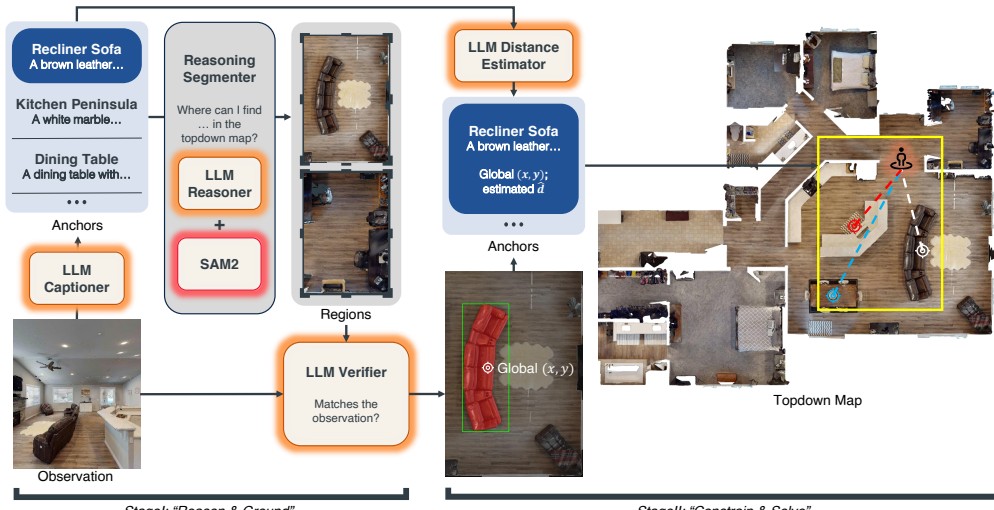

Figure 2: Overview of GeoReasoning. Stage I ("Reason & Ground") extracts map-level semantics and object anchors from an egocentric RGB view, grounds them on the floor map via a promptable segmenter, and verifies cross-view consistency. Stage II ("Constrain & Solve") refines anchor masks, estimates camera-to-object distances, and performs robust trilateration to recover the observer's pose. The pipeline separates high-level reasoning from geometric computation, producing training-free, RGB-only localization with interpretable rationale and uncertainty.

## 2.3 LLM Applications in Embodied Tasks

LLMs increasingly serve as high-level planners for embodied agents, converting language goals into executable actions (e.g., PaLM-SayCan) or reasoning over visual observations in multimodal form (PaLM-E) (Ahn et al., 2022; Driess et al., 2023). VLMaps fuses pretrained vision–language features into 3D reconstructions for text-based navigation, while TagMap replaces dense embeddings with an explicit text-annotated map that is lightweight and LLM-friendly (Huang et al., 2023; Zhang et al., 2024). FLAME (Xu et al., 2024) and NaviLLM (Zheng et al., 2024) further demonstrate generalist LLM-based navigation in urban and indoor settings, highlighting the potential of zero-shot, reasoning-driven embodied agents.

**GeoReasoning** differs by treating self-localization itself as a reasoning problem: aligning a single egocentric RGB view to a 2D floor plan through structured test-time reasoning. This provides a concrete pose and an interpretable rationale in a zero-shot regime, addressing the missing capability of *map-reading localization* for sensor-light embodied AI.

## 3 Method: Structured Reasoning with Anchors and Verification

We cast localization as *reasoning over semantics to deduce geometry*. Given an egocentric RGB observation $I \in \mathbb{R}^{H \times W \times 3}$ and a top–down floor map $M$ (binary or semantic), our goal is to estimate the camera position $p^\star \in \Omega \subset \mathbb{R}^2$ (and optionally yaw $\psi$) in a training-free, RGB-only regime. The central idea is to separate *what matters*—landmarks that are stable and map-visible—from *how to solve*—a small, well-posed geometric problem. This yields a two-stage framework that mirrors human behavior: first reason about salient anchors and ground them on the map, then convert those semantics into metric constraints and solve for the pose. This division of labor of GeoReasoning thus contributes a high-level language reasoning for proposing and verifying anchors, while classical geometry delivers precise, calibrated localization.

**Stage I: Reason & Ground (Object-Centric Anchors).** Starting from $I$, a multimodal LLM $G$ produces overall map description $D$ and a compact set of map-visible *anchors* with textual attributes and coarse geometry,

$$\mathcal{C} = \left\{ (\ell_k, \ \pi_k) \right\}_{k=1}^{K}, \ell_k : \text{category/instance text}, \pi_k : \text{priors (e.g., shape, adjacency)}.$$

Rather than treating perception as low-level appearance matching, $G$ explicitly reasons about *which* objects are distinctive at map scale (e.g., "L-shaped sofa near a corridor end") and *why* they disambiguate symmetric rooms (layout and relational cues). These structured proposals for *anchors* are

then grounded on $M$ by prompting a segmenter $S$ (promptable SAM-style) with text-to-region hints derived from $(\ell_k, \pi_k)$, yielding masks $R_k = S(M; \ell_k, \pi_k)$ and centroid points $a_k \in \mathbb{R}^2$ for $R_k$'s bounding box $\mathcal{N}(R_k)$.

Indoor layouts routinely contain repeated categories (e.g., multiple bathrooms, beds, or tables), and prompt-based segmentation on $M$ may highlight only one instance among several plausible candidates. Moreover, partial views in $I$ often reveal only a subset of an object's footprint, making naive instance matching brittle. To disambiguate these cases while preserving zero-shot generality, we introduce a cross-view semantic *verifier* that operates as a hard gate: it answers whether the egocentric evidence and a candidate map neighborhood plausibly correspond to the *same* object instance and local layout.

To suppress spurious anchors without sacrificing zero-shot generality, we perform cross-view semantic verification using a binary gate

$$b_k = \mathbb{V}\big(I, \operatorname{crop}(M, R_k); \ell_k, \pi_k, D\big) \in \{0, 1\},$$

obtained via a lightweight yes/no query to $G$. Here, the verifier $\mathbb{V}$ takes the egocentric image $I$, a cropped map region around the proposed mask $R_k$, and the anchor's textual priors $(\ell_k, \pi_k)$, and returns *true* if they refer to the same instance (consistent shape, adjacency, and local layout cues) and *false* otherwise. We retain only regions with $b_k = 1$, yielding

$$\mathcal{A} = \big\{(a_k, R_k, \ell_k, b_k)\big\}_{k \in \mathcal{K}}, \quad \mathcal{K} = \{k : b_k = 1\}.$$

This hard-gating step prevents error compounding from mis-grounded duplicates while keeping the pipeline training-free.

**Stage II: Constrain & Solve (Geometric Trilateration).** Stage II starts *from the verified regions of Stage I* and purposefully "zooms in" to refine geometry before solving for pose. For each $(a_k, R_k, \ell_k, b_k) \in \mathcal{A}$, we crop a padded neighborhood of the map, $M_k^{\text{zoom}} = \operatorname{crop}\big(M, \mathcal{N}(R_k)\big)$, and re-run a *local* segmentation $S_{\text{ref}}$ conditioned on $\ell_k$ to obtain a sharper mask $\widehat{R}_k = S_{\text{ref}}(M_k^{\text{zoom}}; \ell_k)$ and a refined anchor $\widehat{a}_k$ (e.g., centroid or a medial-axis mode). The zoom-in step leverages locality and reduces background clutter, yielding more accurate footprints and thus tighter geometric constraints.

At this refined scale, we next obtain a *metric* cue by querying $G$ to estimate the camera–to–object distance in meters, but now with the object identity fixed and the spatial context disambiguated by $\widehat{R}_k$. Denote this scalar by

$$\widehat{d}_k = \mathbb{D}\big(I, \ell_k\big) \in \mathbb{R}_{>0},$$

where $\mathbb{D}$ serves as a zero-shot distance estimator that exploits semantic size priors and perspective cues in $I$. Let $s$ be the pixels-per-meter scale of $M$; each anchor induces a circle constraint with radius $\widehat{\rho}_k = s\,\widehat{d}_k$ centered at $\widehat{a}_k$.

We localize by solving a robust trilateration over the navigable map domain $\Omega \subset \mathbb{R}^2$. Given Stage I–admitted anchors $\{\widehat{a}_k\}_{k:b_k=1}$ and their refined ranges $\{\widehat{\rho}_k\}$, we define the loss $L(p)$ estimate

$$p^\star \in \arg\min_{p \in \Omega} \; L(p) = \sum_{k:\,b_k=1} w_k\, \varphi\Big( \underbrace{\|p - \widehat{a}_k\|_2 - \widehat{\rho}_k}_{r_k(p)} \Big), \tag{1}$$

where $w_k \geq 0$ encodes anchor confidence (e.g., detection score or range variance) and $\varphi$ is a robust penalty (Huber or Tukey) to downweight outliers and mild segmentation–footprint drift. In typical scenes, we set $w_k = 1$ and three non-collinear anchors uniquely determine $p^\star$; additional anchors over-determine equation 1 and improve stability. When residuals cannot be driven below a tolerance, we report calibrated uncertainty as the sublevel set

$$\mathcal{U}_\delta = \{\, p \in \Omega : \; L(p) \leq \delta \,\},$$

visualized either by its local covariance ellipse from $\big(J^\top W J\big)^{-1}$ at $p^\star$ or, for maps, by the minimum-area axis-aligned square enclosing $\mathcal{U}_\delta$ (side length capped at $0.5\,\text{m}$). Degenerate cases (e.g., two anchors in 2D) naturally yield a region rather than a unique point. We use region centriods for further analysis.

This zoom–refine–trilateration flow preserves human-like chain of thoughts: *reasoning first, geometry last*. Stage I uses language to ground *which* regions matter on the map and to enforce cross-view consistency at the instance level; Stage II focuses computation within those verified neighborhoods, re-segments for geometric fidelity, queries a single metric scalar per anchor, and solves a transparent, low-dimensional trilateration. The result is a training-free, RGB-only pipeline that remains robust to appearance shift, clutter, and symmetry while producing either a precise pose or an honest uncertainty region.

**Reasoning-prompted segmentation (RPS).** Given an RGB image $M$ and a natural-language target $(\ell_k, \pi_k)$ (often an implicit, reasoning-heavy description), the module performs *reasoning then masking*. A multimodal LLM $G$ first interprets the instructional prompt $P$ containing information from $(\ell_k, \pi_k)$ and produces (i) a short chain-of-thought (CoT) rationale and (ii) positional prompts consisting of a bounding box $B = [x_1, y_1, x_2, y_2]$ and two interior points $P_1, P_2$ that lie well inside the object. A frozen segmentation backend $\mathcal{F}_{\text{seg}}$ (SAM2 (Ravi et al., 2025)) then consumes $(B, P_1, P_2)$ to generate a mask $R$:

$$(B, P_1, P_2, \text{CoT}) = S_{G(\text{reason})}(M; \ell_k, \pi_k), \qquad R = S_{\mathcal{F}(\text{seg})}(M; B, P_1, P_2).$$

This combination preserves the pixel fidelity of a strong segmenter while letting the MLLM spend capacity on language understanding, disambiguation, and map-level or commonsense reasoning before emitting prompts that are easy for the segmenter to act on. At inference, a single structured instruction (with one in-context example) reliably elicits the json structure. Empirically, strict format rewards aid OOD robustness, and sample-count during RL improves exploration and final accuracy. As a self-contained component, Seg-Zero exposes a narrow interface—$(I, T) \rightarrow (M, B, P1, P2, \text{CoT})$—that plays well with GeoReasoning framework. The CoT trace can be logged for verifiability or fed into downstream reasoning (e.g., consistency checks across views), while $B, P_1, P_2$ remain model-agnostic prompts for any SAM-family decoder.

## 4 EXPERIMENTS

### 4.1 DATASET: CoT-SEG SCENES, VIEWS, AND MAP CONTEXT

**Overview.** **Loc-Bench** is a benchmark built in Habitat–Sim (Savva et al., 2019) on HM3D (Ramakrishnan et al., 2021). Each episode mounts co–located pinhole sensors at $1.6325\,\text{m}$ with $512 \times 512$ resolution: RGB (color_sensor), depth (depth_sensor), and semantic (semantic_sensor). For every scene we sample $3 - 15$ navigable locations from the navmesh depending on the scene size, and capture a *primary* first–person RGB view (the one used in all main experiments) plus three additional rotations at yaw angles $90°, 180°, 270°$ recorded as a backup set. Consequently, the corpus contains 9251 *primary* observations and 37004 total observations across 900 scenes; unless stated otherwise, all reported metrics use only the primary view at each location. We also export a global top–down floorplan used by our methods and baselines, while an agent–overlaid top–down RGB and per–floor occupancy raster are provided solely for visualization and evaluation utilities.

**Data collection and annotations.** Sampling enforces a $10\,\text{cm}$ margin to the nearest obstacle to avoid degenerate wall–dominated frames. For every observation we log camera pose $(x, y, z)$, yaw (deg/rad), floor index, and map scale (pixels–per–meter). *Object ground truth* comprises: (i) instance masks and categories from the simulator's semantic renderings; (ii) short, discriminative instance–level descriptions produced by a state–of–the–art captioner, Qwen2.5–VL (Bai et al., 2025) finetuned on Ego4D (Grauman et al., 2022); and (iii) metric *object distances* computed from the depth sensor by taking the median depth within each instance mask and back–projecting with known intrinsics. The exported top–down floorplan is the only map artifact used by default in experiments; the per–floor occupancy map serves as an *evaluation–time validity test* to reject predictions that fall in non–navigable cells. Agent–overlay maps (top–down RGB with camera icons) are provided for visual reference only.

In realistic indoor layouts a single forward–facing frame can occasionally lack informative cues (e.g., facing a blank wall). We therefore record three orthogonal backups at the same location to guarantee that at least one view is informative; however, to maintain a strict and reproducible protocol, *all primary results are computed on the first view only*, mirroring human "first glance"

usage and ensuring comparability across methods. Backup views are used exclusively for robustness ablations. This design (i) decouples appearance from position via controlled multi–view at a fixed center, (ii) enables geometry–aware checks through the floorplan and the occupancy–based validity gate (evaluation–time only), and (iii) provides metric supervision via depth–derived object ranges, while centering single–view performance.

## 4.2 EVALUATION

We score methods on the *primary* view (first forward-facing frame) at each sampled location. Given a single RGB and the exported top-down floorplan, a method outputs (i) detected objects with categories (and masks if available), (ii) a per-object metric distance estimate, and (iii) optionally a 2D map coordinate (or region centroid) in floorplan space. Evaluation is performed in map space using the known pixel–meter scale, with an *occupancy validity* gate applied to check whether predicted coordinates lie on navigable cells. We then report three families of metrics:

**(A) Object identification.** We match predictions to ground-truth instances by category; a prediction is a *hit* if its category appears among ground-truth objects in the frame. We report the *hit rate* **Obj-ID**.

**(B) Distance estimation.** For every matched object, we compare the predicted range $\hat{d}$ to the ground-truth metric distance $d^\star$ computed from the depth sensor (median depth within the instance mask, back-projected with known intrinsics). We report **R-MAE** $|\hat{d} - d^\star|$.

**(C) Localization / task achievement.** When a method outputs a 2D map point (or region centroid) $\hat{p}$ for an object, we first apply an *occupancy validity* check using the per-floor map (prediction must lie on a navigable coorinates). We then measure the Euclidean localization error $\|\hat{p} - p^\star\|_2$ in meters (where $p^\star$ is the ground-truth map coordinate of the object) and report: Valid-Prediction Rate **VPR**, Mean/Median Localization Error **LE mean/med**, and **S@$r$** for $r \in \{0.1, 0.5, 1, 3\}$ meters (percentage of ground-truth targets whose distance to $\hat{p}$ is $\leq r$). If a region is predicted, $\hat{p}$ is taken as the region centroid.

To contextualize task difficulty, we report two references: (i) *Random*—sampling a coordinate on the map; and (ii) *Human performance*—five annotators localizing 100 queries across 35 scenes. Then we compare against a *Zero-shot MLLM Prompt* baseline that directly instructs a strong multimodal reasoning model to localize from the first-person RGB and the scene floorplan (returning an object list, distances, and a coordinate in the floorplan frame).

## 4.3 QUANTITATIVE RESULTS

We evaluate a representative set of state-of-the-art multimodal LLMs-O3 (OpenAI, 2025), GEMINI-2.5-PRO (Comanici et al., 2025), GPT-4O (Hurst et al., 2024), LLAMA-4-MAVERICK (Meta, 2025), INTERVL3.5 (Chen et al., 2024), LLAVA (Liu et al., 2023), QWEN-2.5-VL (Bai et al., 2025) and QWEN-2.5-OMNI Xu et al. (2025)-covering both proprietary (API) and open-source families. Unless otherwise noted, all scores are computed on the *primary* view (the first forward-facing observation) at each location on Loc-Bench; optional rotations are used only in an ablation.

From Table 1, random sampling produces near-zero accuracy and the average distance is far from ground truth, which confirms that non-trivial structure is required to localize. Human annotators achieve near-perfect meter-level accuracy showing that indicating that a single egocentric frame typically contains enough semantic and geometric cues for reliable global grounding, but S@0.1 remains challenging. The *direct-prompt* baseline also produces unsatisfactory and often near-random localization: models frequently recognize objects and narrate plausible scene context (e.g., "middle of the living room, next to the table") yet output coordinates outside the building or on non–navigable cells. This failure is structural. Direct prompting asks an MLLM to do four difficult tasks *purely in language*: (i) perceive, (ii) describe, (iii) reason over spatial relations, and (iv) transform egocentric descriptions into allocentric map coordinates. As analyzed by Yang et al. (2025), errors in spatial reasoning can be categorized into *visual perception*, *linguistic intelligence*, *relational reasoning*, and *egocentric→allocentric* transformation; in our experiments, the latter two dominate. Two deficits explain the pattern: first, the lack of *pixel–level grounding* leaves textual anchors ("left of the sofa", "near the corridor end") unbound to concrete regions on the floorplan; second, the egocentric → allocentric transform is under–specified by language alone–bearing, scale, and pose priors are not explicitly represented–so small narrative inconsistencies translate into meter–scale coordi-

| | Intermediates | | Localization Result | | | | | | |
|---|---|---|---|---|---|---|---|---|---|
| Model | Obj-ID ↑ | R-MAE ↓ | VPR ↑ | LE mean ↓ | med ↓ | S@0.1 ↑ | S@0.5 ↑ | S@1 ↑ | S@3 ↑ |
| Random | - | - | 0.768 | 5.47 | 4.49 | 0.000 | 0.003 | 0.030 | 0.104 |
| Human Performance | 0.99 | 0.25 | 1.000 | 0.30 | 0.28 | 0.156 | 0.783 | 0.970 | 0.973 |
| Baseline (o3) | 0.94 | 1.58 | 0.754 | 4.68 | 3.09 | 0.001 | 0.011 | 0.085 | 0.102 |
| Qwen2.5-Omni-3B | 0.82 | 1.03 | 0.812 | 1.69 | 1.43 | 0.012 | 0.257 | 0.470 | 0.631 |
| Qwen2.5-Omni-7B | 0.86 | 0.82 | 0.878 | 1.58 | 1.37 | 0.009 | 0.279 | 0.467 | 0.657 |
| Qwen2.5-VL-7B | 0.87 | 0.81 | 0.878 | 1.55 | 1.35 | 0.021 | 0.282 | 0.481 | 0.701 |
| InternVL3.5-8B | 0.86 | 0.92 | 0.862 | 1.68 | 1.42 | 0.019 | 0.279 | 0.415 | 0.657 |
| LLaVA-v1.5-7B | 0.82 | 1.21 | 0.822 | 1.75 | 1.58 | 0.049 | 0.233 | 0.396 | 0.598 |
| Llama-4-Scout | 0.91 | 0.75 | 0.876 | 1.04 | 0.91 | 0.049 | 0.479 | 0.627 | 0.728 |
| Llama-4-Maverick | 0.91 | 0.77 | 0.898 | 0.94 | 0.85 | 0.055 | 0.481 | 0.659 | 0.749 |
| GPT-4o | 0.92 | 0.69 | 0.923 | 0.85 | 0.76 | 0.051 | 0.489 | 0.692 | 0.763 |
| Gemini 2.5 Pro | 0.95 | **0.46** | **0.971** | 0.48 | 0.45 | **0.072** | 0.554 | 0.751 | 0.858 |
| o3 | **0.96** | 0.48 | 0.965 | **0.46** | **0.45** | 0.068 | **0.568** | **0.782** | **0.891** |

Table 1: Primary-view evaluation (single RGB) on floorplan map space, with notable difference between o3 baseline (top row) and o3 GeoReasoning results (bottom). **Bold** indicates the best results.

nate errors. Without explicit constraints on pixels, or geometry, a fluent chain–of–thought remains linguistically coherent but metrically unconstrained, yielding unreliable coordinates.

Our *GeoReasoning* pipeline addresses these deficits by two-stage-inference. The MLLM proposes discriminative anchors that are *describe-able* and map-visible; the segmenter converts those hypotheses into concrete regions on the floorplan; a verifier enforces cross-view consistency between the proposed region and RGB evidence (texture, shape, adjacency). A practical consequence is that we can reason about-and control-the geometry. After verification, we *zoom in* on the candidate region, which raises segmentation and reduces localization error.

We use the $< 3\,\text{m}$ metric as an indicator for Stage 1 quality. In practice, once the hypothesized region is correctly grounded on the map, the subsequent triangulation almost always yields a final error under $3\,\text{m}$. Thus, improvements on $< 3\,\text{m}$ primarily reflect better hypothesis generation and geometric grounding rather than downstream solver effects.

Proprietary large MLLMs substantially outperform smaller open-source models on Stage I, as observed on the $< 3\,\text{m}$ metric. Notably, object detection scores show no significant difference across models, and the segmenter is frozen in all comparisons. The performance gap therefore arises upstream: *text-level reasoning for segmentation prompts* and *cross-modal alignment* in the validator. Larger models-by virtue of stronger multimodal alignment and inference-time reasoning-produce more discriminative, map-visible anchors and more consistent text↔region matches. Nevertheless, even the smaller open–source models deliver substantial absolute gains over the direct–prompt baseline, underscoring the value of explicit grounding and validator–guided alignment.

During Stage II, distance estimated by MLLM are still noisy. Once Stage I has grounded the correct region, modest errors in MLLM-estimated distances have little effect on trilateration or the selected region as they could be damped by well-spread anchors and over-determination ($k > 3$). Across models, the fraction of *sub-meter* localizations ($\leq 0.5\,\text{m}$) *conditioned on* successful region grounding ($< 3\,\text{m}$) is remarkably stable, indicating that trilateration is not the bottleneck. However, the *ultra-fine* regime ($< 0.1\,\text{m}$) remains challenging, largely due to residual distance estimation noises, even after zoom-in. Overall, these trends support the view that Stage I reasoning and grounding-not the metric solver-govern end-to-end performance.

In failure analyses, the dominant failure modes are (a) *layout aliasing* from highly symmetric floorplans (parallel corridors, grid-like rooms), and (b) *anchor ambiguity* when language selects objects with non-unique map signatures ("a chair near a table"). Our ablations indicate that verification and zoom-in steps are most beneficial in (b), while multi-view backups mainly mitigate (a) by increasing the chance that at least one view sees an asymmetric cue.

## 4.4 QUALITATIVE RESULTS

Figure 3 illustrates that GeoReasoning achieves reliable map-space localization. When different observations contain near-identical object sets, our pipeline first proposes candidate regions via the reasoning segmenter and then uses an LLM verifier to test whether each masked region is consistent with the RGB observation (layout, adjacency, and relational cues). For example, Observations 2 and 3 both include open shelving and a black sofa, leading to the same set of candidates; nevertheless, the verifier reasons over spatial configuration and selects the correct mask (blue-ringed) that we

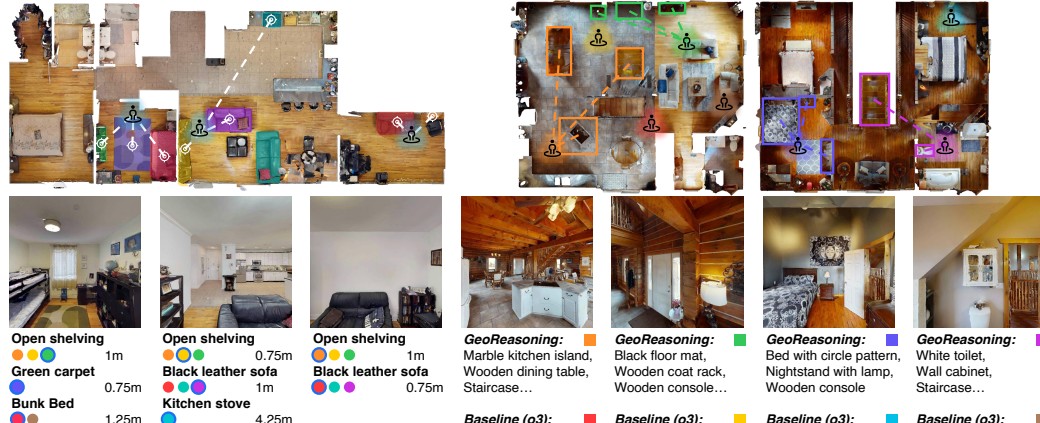

Figure 3: Qualitative evaluation of GeoReasoning. (Left) Despite near-duplicate textual descriptions and visually similar objects, GeoReasoning grounds each anchor to concrete map regions (colored masks). (Right) Across multiple observations in a multi-floor scene, GeoReasoning's predicted locations consistently align with the true region, whereas the baseline behaves close to random. Colors distinguish observations or masks; blue rings mark the selected mask.

subsequently use for trilateration. Such scenarios—multiple beds, tables, or sofas across rooms—are common in indoor environments, and the verification step consistently filters look-alike distractors, evidencing robustness.

We compare GeoReasoning to the baseline (o3) by visualizing their predictions. Directly aligning egocentric views to top-down maps with a single, long chain of thought remains a significant challenge for SOTA MLLMs Their spatial reasoning abilities often degrades as the reasoning trace grows longer (Yang et al., 2025). In contrast, GeoReasoning decomposes localization into smaller, structured subproblems with verifiable intermediate artifacts (candidate masks and region checks) that modern MLLMs handle well. As a result, GeoReasoning's enables modern MLLM in performing single-view localization with our factorized, verify-then-localize design.

## 4.5 ABLATION STUDY

We dissect GeoReasoning to attribute where the gains arise.

*Removing cross-view verification (–verifier).* Object detection and distance estimation remain unchanged, but localization collapses: both LE and success rates drop significantly. This highlights the importance of verifier's primary role on instance disambiguation among repeated categories and enforcing scene-context consistency.

*Removing zoom-in refinement (–refinement).* Without refinement, LE(med) increases and success rates decreases modestly. Refinement therefore delivers consistent but moderate gains by sharpening masks and rejecting spurious candidates in clutter, with the largest benefit at accurate thresholds.

*Improving distance estimation (GT and DepthLM).* Replacing LLM-estimated ranges with *ground-truth distances* keeps macrometrics stable but boosts ultra-fine accuracy substantially. A practical middle ground is a text-promptable distance estimator (*DepthLM*), which raises S@0.1 to 0.102. These results show that range quality chiefly affects sub-decimeter success, while the verifier and geometry dominate meter-level accuracy.

*Camera poses and backup rotations ($R = 4$).* Each location provides three auxiliary yaw rotations $(90°, 180°, 270°)$ in addition to the primary view. We adopt a pragmatic policy: require at least $|\mathcal{A}| \geq 3$ verified anchors; if the primary view yields fewer, sequentially query the next rotation(s) until the threshold is met (or $R=4$ is reached). Results in Table 2 show monotonic improvements in VPR, S@0.5/3, and LE as $R$ increases. Gains are largest in hard cases where the primary view faces a blank wall or feature-poor corridor, reflecting the value of *viewpoint diversity* for anchor discovery.

To summarize, the verifier is the principal driver of meter-scale accuracy; refinement provides consistent, tighter-threshold gains; improved ranges chiefly affect the S@0.1 regime; and limited multi-rotation offers a compute-accuracy trade-off that strengthens robustness when the primary view is anchor-poor.

| | Intermediates | | Localization Result | | | | | | | |
|---|---|---|---|---|---|---|---|---|---|---|
| Model | Obj-ID ↑ | R-MAE ↓ | VPR ↑ | LE mean ↓ | med ↓ | S@0.1 ↑ | S@0.5 ↑ | S@1 ↑ | S@3 ↑ |
| Baseline (o3) | 0.94 | 1.58 | 0.754 | 4.68 | 3.09 | 0.001 | 0.011 | 0.085 | 0.102 |
| o3 | 0.96 | 0.48 | 0.965 | 0.46 | 0.45 | 0.068 | 0.568 | 0.782 | 0.891 |
| o3 (w/o verifier) | 0.96 | 0.48 | 0.953 | 1.59 | 1.25 | 0.015 | 0.158 | 0.229 | 0.392 |
| o3 (w/o refinement) | 0.96 | 0.49 | 0.960 | 0.54 | 0.49 | 0.045 | 0.535 | 0.745 | 0.869 |
| o3 (gt distance) | 0.96 | 0.00 | 0.965 | 0.46 | 0.45 | 0.118 | 0.568 | 0.782 | 0.892 |
| o3 (DepthLM) | 0.96 | 0.13 | 0.969 | 0.41 | 0.42 | 0.102 | 0.571 | 0.784 | 0.892 |
| o3 (R=4) | 0.96 | 0.49 | 0.968 | 0.45 | 0.44 | 0.068 | 0.575 | 0.799 | 0.905 |

Table 2: **Module-wise ablations** on Loc-Bench with a fixed o3 backbone. We report metrics for: removal of the cross-view verifier and zoom-in refinement, replacing LLM ranges with GT or *DepthLM*, and multi-rotation ($R{=}4$). Removing the verifier causes the largest meter-level drop; refinement provides consistent gains (notably at S@0.1); better ranges chiefly improve S@0.1; and $R{=}4$ yields consistent improvements.

### 4.6 EXTENDING TO OUTDOOR SCENES

In this section, we demonstrate how **GeoReasoning** naturally extends to outdoor scenes. By treating a satellite BEV image as the "floorplan" and using a single egocentric frame, our zero-shot pipeline identifies map-visible, permanent landmarks (e.g., domes, statues), estimates their ranges, and recovers the camera pose via robust trilateration. With only a minor prompt adjustment ("prefer permanent structures; ignore people/vehicles/vegetation"), the same pipeline produces predictions that fall in the correct region and closely match the ground truth, which demonstrates that our reasoning-plus-geometry paradigm is not limited to indoor layouts.

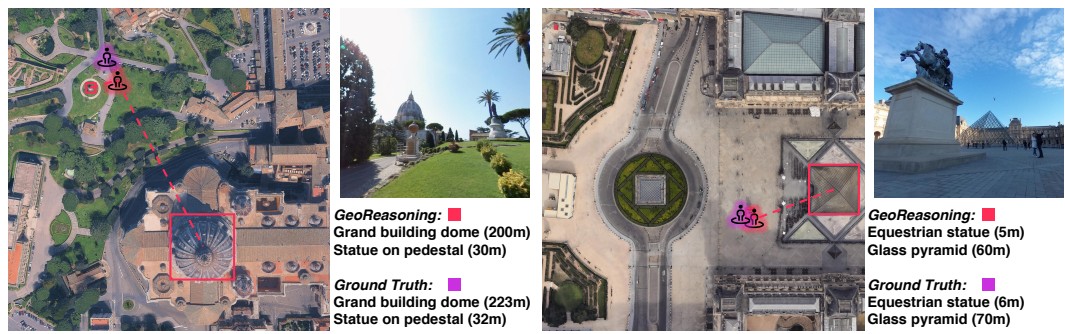

Figure 4: **Outdoor GeoReasoning** can successfully locate agents in Vatican City and Louvre Plaza.

## 5 DISCUSSION

**Limitations and Future Work.** Our current pipeline already closes most of the gap to practical use by emphasizing relational cues ("what is where") over exact metering, but two aspects invite improvement. First, sub-decimeter accuracy ($<0.1$ m) is still hard because monocular, language-driven range estimates remain noisy; even after zoom-in and robust trilateration, the ultra-fine regime is governed by residual distance errors from the MLLM rather than by the solver itself. Second, scenes with strong symmetries or anchors with non-unique map signatures can still alias; here, principled disambiguation could leverage multi-view consistency. Beyond indoor floorplans, the same factorized recipe naturally extends to outdoor malls, campuses, and transit hubs by swapping in style-robust 2D maps and adding modules for multi-floor transitions and dynamic clutter.

**Concluding Remarks.** GeoReasoning reframes localization as "reason first, geometry last," delivering interpretable anchors, verifiable intermediate checks, and robust pose recovery from a single RGB plus a floorplan. On our new benchmark, the factorized design consistently improves map-space grounding over direct prompts—particularly in the success@r metrics that matters for practical navigation—and surfaces crisp research targets for the field. We view this as a step toward human-like map reading: people rarely guess metric distances to 10 cm, yet navigate confidently by structure and relations; likewise, our results show that once the right region is grounded, modest metric errors seldom change the outcome. We hope this structured blend of semantics and geometry will catalyze broader study across domains—indoor, outdoor, and beyond—and inspire compact modules that plug into LLM agents for planning, instruction following, and embodied AI.

**Ethics Statement.** This work studies RGB-to-map localization on public indoor-scene datasets (HM3D/Habitat) and does not involve human subjects, face recognition, voice recordings, or personally identifiable information. All data are used under their respective licenses; no new data are collected. Our method localizes a camera with respect to floorplans and furniture-not people-and we explicitly avoid person-level tracking or re-identification. Potential risks include misuse for covert surveillance or mapping of private spaces. To mitigate this, we (i) restrict experiments to public research datasets, (ii) release code and models strictly for research and educational use, (iii) document intended uses and limitations, and (iv) refrain from releasing any content that could deanonymize private environments. We are unaware of conflicts of interest or sensitive sponsorship. No IRB approval was required as no human-subject research was conducted.

**Reproducibility Statement.** After the paper is published, we will release: (1) full code to run inference and evaluation; (2) exact prompts and output schemas used by all MLLMs; (3) configuration files, random seeds, API settings, and environment specifications (CUDA/driver, Python and package versions); (4) scripts to download and prepare HM3D/Habitat data; and (5) per-scene prediction logs (anchors, masks, ranges, and poses) to verify metrics. The released evaluation script recomputes all tables/figures from logged predictions, enabling end-to-end reproduction from raw data to final results.

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

# A APPENDIX

In the appendix, we present:

- Section A.1: More Qualitative Results
- Section A.2: LLM Usage Statement

## A.1 MORE QUALITATIVE RESULTS

**Temporal Modeling** We illustrate that **GeoReasoning** extends naturally to short videos without any architectural changes. From a clip, we sample a few frames and run the single-frame pipeline on each, yielding per-frame poses. Linking these predictions over time provides continuous tracking. Because adjacent frames share co-visible anchors and the true motion between frames is small, the cross-view verifier resolves instance ambiguities more reliably and the fused estimate suppresses single-frame noise. In practice, this temporal context further improves accuracy and stabilizes trajectories, especially in symmetry-prone layouts. See Fig. 5.

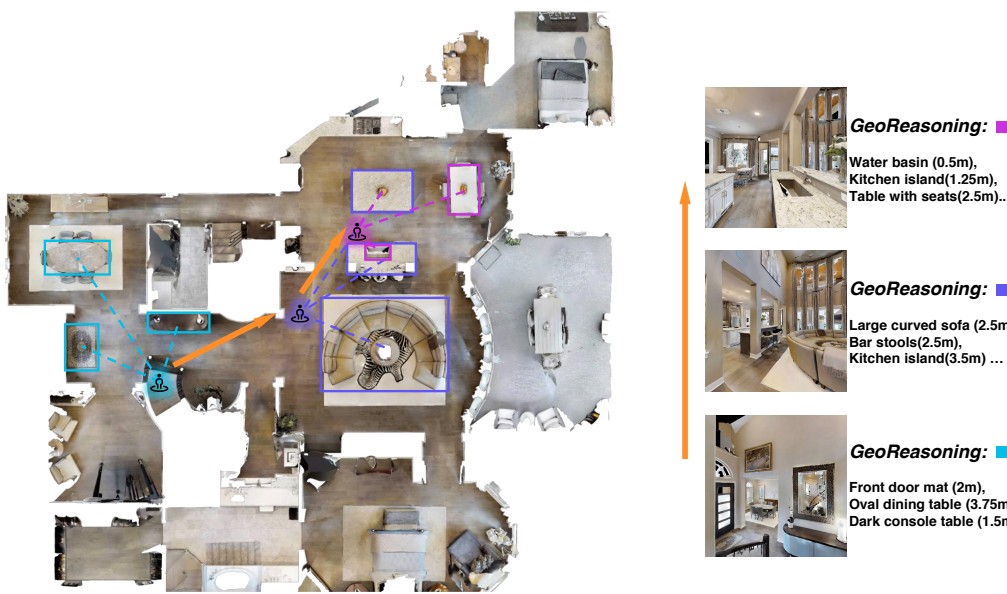

Figure 5: Temporal modeling.

**Failure Case Analysis**  In Fig 6, we demonstrate two failure patterns.

*Rare cases of symmetry*: several hotel rooms share virtually identical layouts and furnishings. Under this symmetry, the LLM verifier sees a region-image pair that is equally consistent with multiple rooms, so no unique assignment exists. With geometric and semantic cues indistinguishable, the evidence is information-theoretically insufficient to pinpoint the correct room-even for humans-making the final selection effectively arbitrary among symmetric alternatives.

*No visual reference*: in other scenes—such as unfurnished sample rooms—there are essentially no distinctive objects or textures that can serve as anchor points. Deprived of recognizable furniture or decor, the LLM captioner resorts to describing generic elements (e.g., "white walls," "wooden floor," "ceiling light") that cannot be uniquely grounded on the top-down map. As a result, the verifier is impossible to reliably assess whether the predicted mask is correct. This lack of discriminative visual evidence again leads to ambiguous, often incorrect localization.

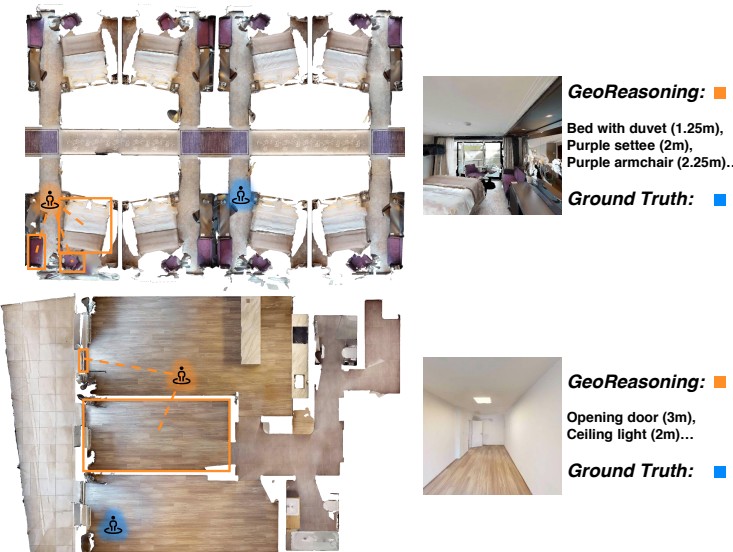

Figure 6: Failure Cases

**More qualitative examples** We provide more qualitative examples to demonstrates **GeoReasoning**'s ability in localizing an observation in cluttered scenes.

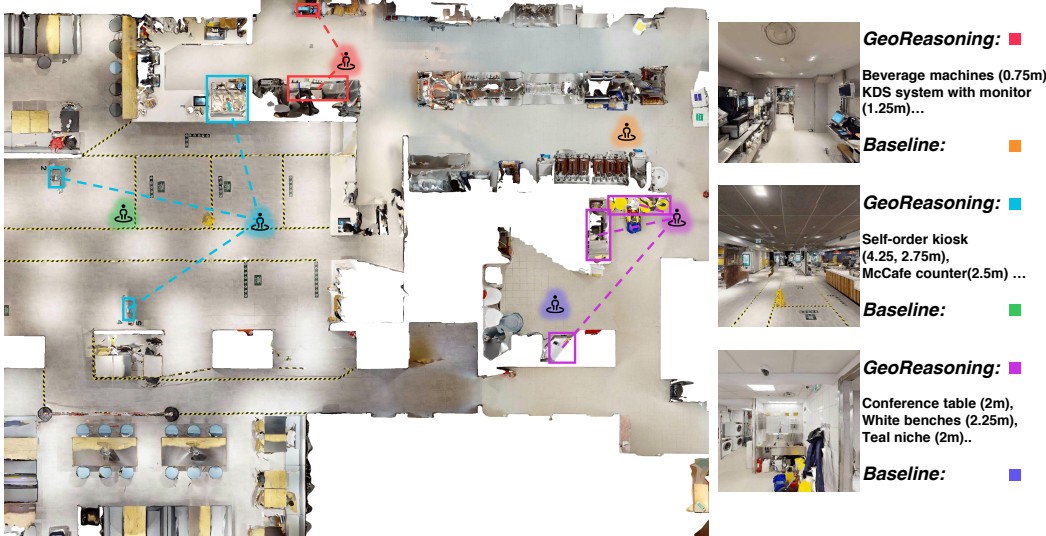

**GeoReasoning:** ■

**Beverage machines (0.75m), KDS system with monitor (1.25m)…**

**Baseline:** ■

**GeoReasoning:** ■

**Self-order kiosk (4.25, 2.75m), McCafe counter(2.5m) …**

**Baseline:** ■

**GeoReasoning:** ■

**Conference table (2m), White benches (2.25m), Teal niche (2m)..**

**Baseline:** ■

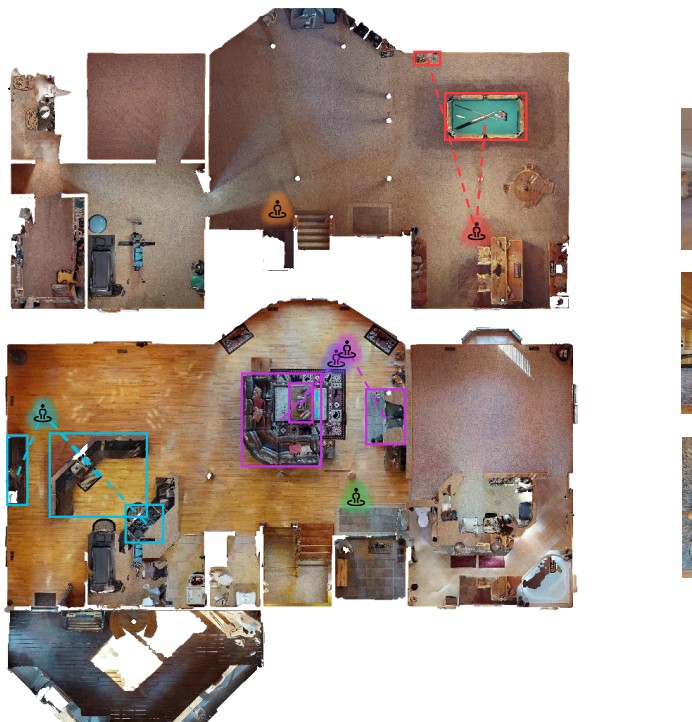

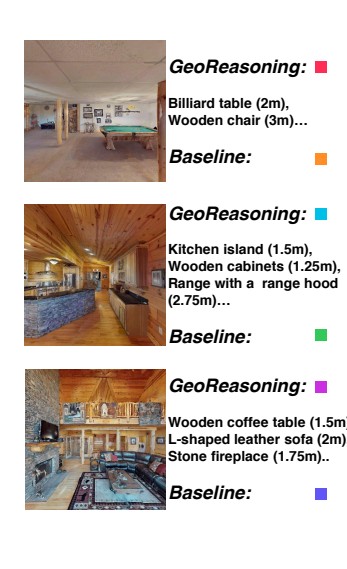

**GeoReasoning:** ■

**Billiard table (2m), Wooden chair (3m)…**

**Baseline:** ■

**GeoReasoning:** ■

**Kitchen island (1.5m), Wooden cabinets (1.25m), Range with a range hood (2.75m)…**

**Baseline:** ■

**GeoReasoning:** ■

**Wooden coffee table (1.5m), L-shaped leather sofa (2m), Stone fireplace (1.75m)..**

**Baseline:** ■

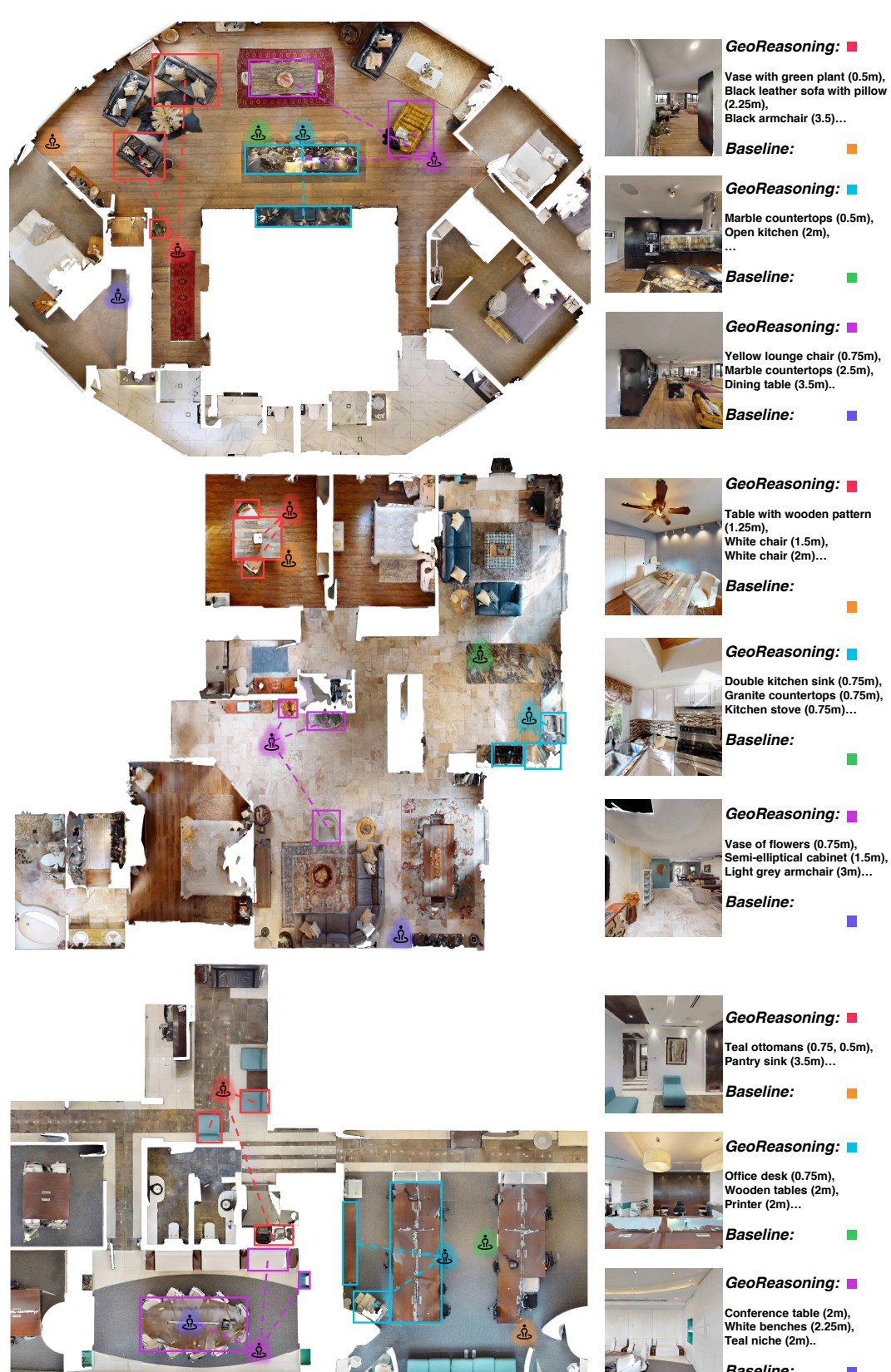

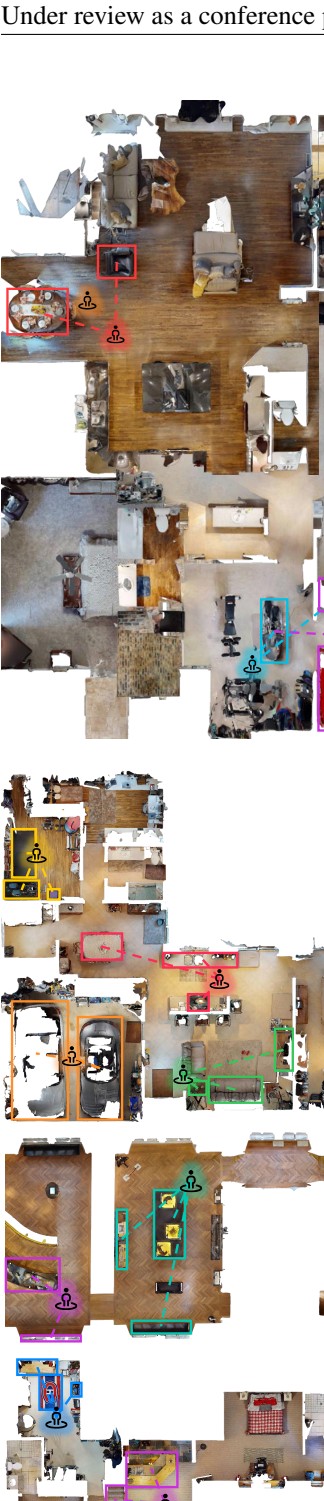
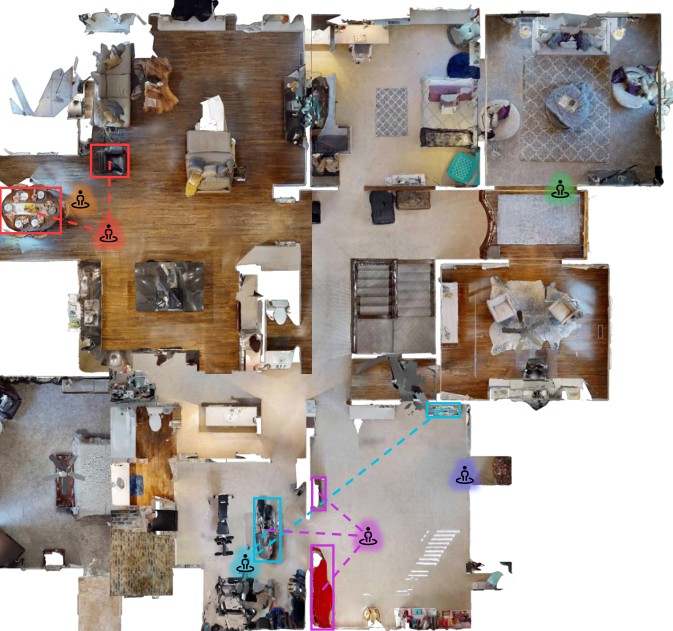
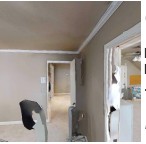
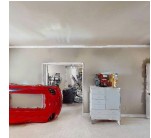

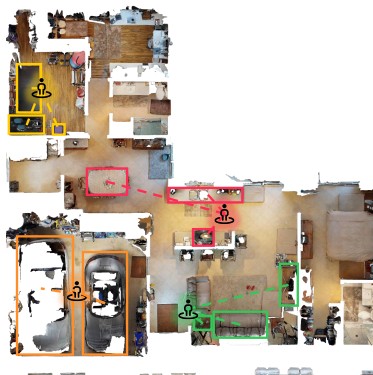

*GeoReasoning:* ■

**Round table(1.75m),
Dark leather armchair (1.5m)
...**

*Baseline:* ■

*GeoReasoning:* ■

**Elliptical machine (0.5m),
Blue blanket (4.5m),
...**

*Baseline:* ■

*GeoReasoning:* ■

**Red car-shaped frame (1.5m),
White cabinet (1.5m),
Elliptical machine (2.5m)..**

*Baseline:* ■

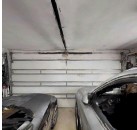
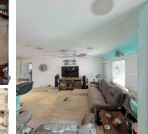
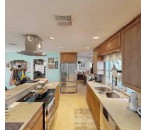

*GeoReasoning:* ■

**Elliptical cross-trainer (1m),
Purple step stool (1.25m),
Yoga mat (0.25m)...**

*GeoReasoning:* ■

**Dark grey sports cat (0.75m),
Light grey SUV (0.75m),
Garage door (1.75m)...**

*GeoReasoning:* ■

**TV on media console (3m),
Grey Sofa (1.5m),
Coffee Table (0.5m)...**

*GeoReasoning:* ■

**Open kitchen with water sink
(0.75m),
Black oven with stove
(0.75m),
Dining table (3.5m)...**

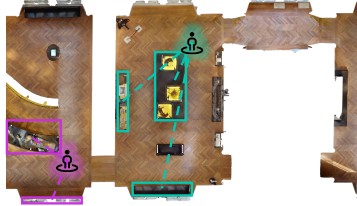
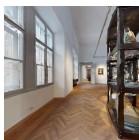

*GeoReasoning:* ■

**Display stand with
instruments (1.5m),
Display cabinets (3m),
Wooden bench (4.5m)...**

*GeoReasoning:* ■

**Curved display unit (2m),
Tall windows (1.5m)...**

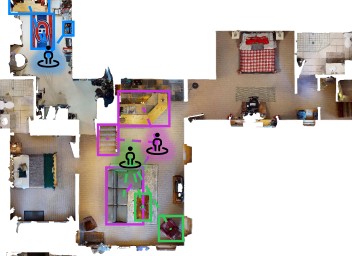
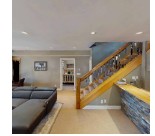

*GeoReasoning:* ■

**Toolbox (1.25m),
Workbench (1.5m),
Floor runner rug with logo
(0.75m)...**

*GeoReasoning:* ■

**Sectional leather sofa (2m),
Wooden staircase (2m),
Wooden bar/ledge (1m)...**

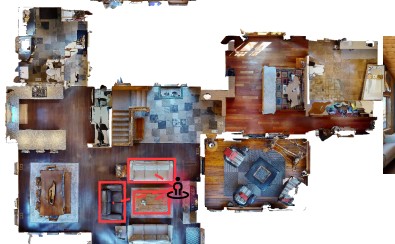
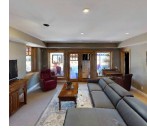

*GeoReasoning:* ■

**Wooden coffee table (0.5m),
White sofa (0.75m),
Black sofa (2m)...**

*GeoReasoning:* ■

**L-shaped sofa (1.5m),
Wooden coffee table (1.75m),
Red armchair (3.25m)...**

## A.2 LLM USAGE

We used an LLM strictly as a general-purpose writing assistant for grammar polishing, wording suggestions, and minor edits to improve clarity and flow. The LLM did not contribute to research ideation, experimental design, implementation, analysis, or result interpretation. All scientific claims, equations, and conclusions were authored and verified by the human authors. The authors take full responsibility for all content and ensure that no material produced with LLM assistance constitutes plagiarism or scientific misconduct. The LLM is not listed as an author.

