# OpenReview forum: "GeoReasoning: Structured Semantic Reasoning for Image-to-Map Localization"
_ICLR.cc/2026/Conference — Submitted to ICLR 2026_

### Official Review · Reviewer_YuzH · 2025-10-29

**Soundness:** 3
**Presentation:** 3
**Contribution:** 2
**Rating:** 2
**Confidence:** 4

**Summary:**

This paper introduces GeoReasoning, a training-free framework for indoor reasoning localization that leverages large language models. The authors also present a novel indoor benchmark specifically designed for this task. The core of the proposed solution is a zero-shot framework that decomposes the complex localization problem into two stages: structured semantic reasoning followed by geometric verification. This approach explicitly models anchor (landmark) selection, verifies cross-view consistency to handle ambiguity, and uses robust trilateration to solve for the final pose. As demonstrated in the evaluations, the GeoReasoning framework significantly outperforms direct-prediction baselines, which often fail at this task.

**Strengths:**

1.	Overall, The paper is well-written, clearly articulating the reasoning localization concept. The proposed GeoReasoning framework is presented logically, and its two-stage (Reason & Ground, Constrain & Solve) methodology is easy to follow.
2.	A key contribution is the proposal of a new indoor localization benchmark, which is derived from an existing dataset. The experimental validation further confirms that the proposed method achieves better performance over the baselines

**Weaknesses:**

1. The claimed reasoning localization paradigm is said to move beyond geometry-first methods. However, it still appears to be closely related to retrieval-based localization. The system essentially searches for geometric landmarks, employs large language models and segmentation to identify potential landmarks, expands the candidate set, and then applies additional rule-based reasoning (e.g., object associations) to refine localization.
2. The authors argue that traditional SLAM-based localization pipelines, while effective, suffer from sensor noise, calibration drift, dynamic scenes, and appearance changes, and require careful tuning for long-term deployment. However, I do not find convincing evidence in this paper that the proposed method actually mitigates these issues. In fact, similar performance improvements can also be achieved by simply expanding the candidate set and introducing additional rules, such as re-ranking strategies.
3. The experiments only compare different language models within the proposed framework, without benchmarking against other representative localization approaches such as LalaLoc or SceneGraphLoc[1] and etc. As a result, it is difficult to substantiate the claimed advantages of this “new paradigm.”
4. The method is evaluated on a newly introduced dataset rather than on existing benchmarks. This raises the concern that the approach may rely on specific data conditions or annotations, which could limit its general applicability.
5. The authors further claim that the proposed method generalizes well to outdoor environments. I remain skeptical of this assertion, as no experimental evidence or quantitative validation is provided to support it.

[1] Scenegraphloc: Cross-modal coarse visual localization on 3d scene graphs. ECCV2024.

**Questions:**

please seek weaknesses.

---

> ### Author Response · Authors · 2025-11-24
>
> > ### W1 - Comparison
>
> We agree that localization cannot be divorced from geometry. The distinction is how geometry is used. Retrieval-style methods (e.g., feature matching, room/scene-graph retrieval) hinge on query–gallery correspondence: they compare the current view to a prebuilt feature database or visual/graph descriptors and return the nearest pose(s). By contrast, GeoReasoning performs generating and testing map-space hypotheses on the fly using semantic reasoning plus geometry, with verifiable intermediates (anchors, masks, ranges). The "rules" are not strict matching: the segmenter and verifier’s judgments come from the MLLM’s relational reasoning, and the final pose is obtained by robust trilateration.
>
> The "new paradigm" claim is not about us inventing correlation. Rather it refers to our novel approach that test-time factorization with multimodal LLMs, without task-specific training or galleries, already deliver global floorplan localization from a single view with interpretable reasoning. This constitutes a distinct, forward-looking pipeline and, in practice, is attractive for unseen buildings, rapidly changing environments, and low-overhead deployments where building and maintaining retrieval bases or room-feature maps is impractical.
>
> > ### W2/3 - Comparison with traditional localization methods
>
> LaLaLoc (wall/room structure, panoramic input). It infers room-scale wall layouts from panoramas and matches against precomputed room/structural features. Its reliance on room-structure cues makes it fail to handle large, open, or outdoor-like areas where wall inference is ambiguous.
>
> SceneGraphLoc (retrieval over 3D scene graphs). It builds a (2D/3D) scene graph of objects and relations and searches a pre-built gallery of geo-referenced scene graphs for the nearest match. In practice, this requires a large, curated retrieval base per environment and is highly sensitive to scene changes—even modest furniture rearrangements or seasonal decor can degrade retrieval. Building such a base is already non-trivial (capture, reconstruction/graphing, geo-referencing), and keeping it fresh demands continuous re-capture, re-graph construction, and re-indexing, which is time-consuming and operationally heavy.
>
> GeoReasoning’s goal is training-free, single-image global localization on a provided floorplan—returning a coordinate in the map frame with verifiable intermediates (anchors, masks, distances). The pipeline requires no precomputed room/3D features, no gallery, no odometry/calibration, and it degrades gracefully under modest map drift by (i) preferring persistent structural anchors, (ii) verifying egocentric with map consistency, and (iii) solving pose with robust trilateration.
>
> To run LaLaLoc/SceneGraphLoc on Loc-Bench would require substantial adaptation, which would either change the task (no longer single-view, floorplan-framed) or privilege assumptions our benchmark deliberately avoids; conversely, forcing those systems into our constraints would artificially cripple them. We therefore opt for a clear table, and we strengthen our internal evidence with fine-grained ablations (Sec 4.5 Table 2) showing which components drive GeoReasoning’s gains.
>
> > ### W4 - Dataset
>
> Our task—reasoning localization from a single RGB image to a 2D floorplan with interpretable, step-wise reasoning—does not align with existing SLAM/re-localization benchmarks, which assume different inputs (e.g., LiDAR/odometry, SfM maps) and objectives (pose tracking or 3D re-localization), making direct use or fair comparison impossible. We therefore built Loc-Bench explicitly for image-to-map grounding while keeping it training-free and sensor-agnostic. The method, GeoReasoning, runs zero-shot without environment-specific training, which reduces the risk of overfitting to any dataset idiosyncrasies.
>
> Loc-Bench is broad and protocolized, not tailored. Loc-Bench is constructed on top of standard, public assets (HM3D within Habitat-Sim) and includes 900 scenes with 9,251 primary observations (and 37,004 total with backups), covering diverse layouts and viewpoints. Sampling enforces geometric sanity (e.g., safe margins to obstacles), logs pose and map scale, and exports both semantic and occupancy-based maps. These design choices aim for breadth and reproducibility rather than dataset-specific shortcuts.
>
> We commit to releasing complete code, exact prompts and output schemas for all MLLMs, scripts to fetch HM3D/Habitat data, and per-scene evaluation logs (anchors, masks, ranges, poses) so others can recompute every table/figure end-to-end. The ethics statement clarifies that we use only public indoor-scene datasets and collect no new personal data. Together, these steps enable independent verification and extension of our benchmark to additional settings.
>
> > ### W5 - Outdoor
>
> We provided a proof-of-concept example (Sec 4.6) illustrating that GeoReasoning is not restricted to indoor layouts and can be extended to outdoor scenes.

---

### Official Review · Reviewer_PKvn · 2025-10-31

**Soundness:** 3
**Presentation:** 2
**Contribution:** 2
**Rating:** 2
**Confidence:** 4

**Summary:**

The authors propose a novel localization paradigm, which leverages the reasoning power of MLLMs to handle indoor-localization tasks. Without training, it decomposes the indoor-localization into the semantic reasoning and geometric trilateration stages. A new benchmark is released for this reasoning localization paradigm.  Compared with the direct-prompt baseline, the proposed GeoReasoning gains obvious improvements in the zero-shot setting.

**Strengths:**

The proposed reasoning localization framework contains two stages. The first extract localization anchors (landmarks) through MLLM captioning and segmenter prompting, and it is more time&memory efficient compared to the sophisticated retrieval used in relocalization and SLAM systems. The second stage constrains the reasoning results with geometric trilateration, which prevents fragile matching and makes it more robust when facing scene clutter, repetition and symmetry.

**Weaknesses:**

The application of the proposed egocentric-floor localization is limited: indoor objects are often moved or replaced in daily life. The framework localizes itself by reasoning and triangulating objects both observed in the egocentric image and the floor plan,  which is unreliable compared with matching (contains more anchor points and excludes irrelevant parts from RANSAC). Moreover,  it is hard for users to update the floor plan without other mapping or rendering techniques, but users can easily update scenes by uploading new egocentric images in traditional localization pipelines.

The proposed method lacks comparison with traditional localization and direct-prompt methods, partly due to the egocentric-floor localization settings, which hinder the demonstration of the contributions mentioned in Strengths.

**Questions:**

Distance reasoning from a single image is inaccurate for SOTA MLLMs, which would be the bottleneck of the localization error under S@1. The authors could leverage monocular depth estimators like Moge-2 or more related DepthLM to get more accurate depth(and camera model) priors.

It would be better to add more visual analysis for failure cases in Line 432-436, such as a pie chart.

Equations are not strictly labeled.  In equations under Line 283，$T$ and $R$ are not mentioned before, and $M$ is duplicated with the symbol of the RGB image.

---

> ### Author Response · Authors · 2025-11-24
>
> > ### W1 - Dynamic Scene
>
> Our segmenter-verifier pipeline is deliberately biased toward persistent, high-salience anchors (structural cues and typically stationary large items such as built-ins, beds, sofas, tables). In daily indoor settings these anchors rarely move, and the cross-view semantic verifier further relies on unchanged contextual cues (e.g., doorways, corridor turns, fixed cabinetry) to validate or reject a candidate region. When the map is outdated or slightly miscalibrated, the system degrades gracefully: verifier scores drop but the correct mask still ranks high.
>
> By contrast, the claim that "users can easily update scenes by uploading new egocentric images" only holds for retrieval-based pipelines; it does not update geometry or navigability—it merely enlarges a gallery and typically still requires SfM/pose labeling, wide coverage, and re-indexing. Moreover, feature-matching methods (e.g., SceneGraphLoc) are sensitive to precomputed room/3D features and often need re-mapping even when slightest furnishings change or when scale/registration drifts. Our semantic + geometric factorization avoids dense appearance correspondences; it prefers fixed structure and compensates for small scale/rotation errors—yielding robustness under the very conditions where feature-matching approaches are most brittle.
>
> Finally, if a true map refresh is desired, modern sparse-to-3D reconstruction methods (e.g., VGGT and other feedforward networks) can quickly reconstruct a lightweight 3D and produce an updated top-down floorplan in certain area—but that is beyond the scope of the paper. Our goal here is training-free global localization that remains usable under modest map drift, which our segmenter-verifier-geometry design achieves in practice.
>
>
> > ### W2 - Comparison
>
> We respectfully clarify that (1) we do compare against direct-prompt baselines (Table 1), and (2) comparisons with traditional supervised methods are not feasible due to fundamental differences in problem definition (zero-shot/single-view vs. supervised/panoramic/retrieval-based).
>
> We acknowledge that traditional methods exist, but they operate under fundamentally different constraints that make direct comparison impossible without altering the task entirely:
>
> Methods like LaLaLoc typically rely on panoramic (360°) images to infer room layouts. Our benchmark (Loc-Bench) and method target the more challenging and realistic single-perspective (limited FOV) setting. Applying panorama-based methods to limited-FOV images leads to degenerate wall inference, while upgrading our input to panoramas would violate the constraints of standard robot/wearable camera setups.
>
> Retrieval-based methods (e.g., SceneGraphLoc) require pre-building extensive databases (3D scene graphs) and training on environment-specific data. In contrast, GeoReasoning is zero-shot and training-free, requiring only a standard 2D floor plan.
>
> The reviewer noted that the setting might "hinder the demonstration of contributions." We argue the opposite: the egocentric-to-floorplan setting is precisely what highlights our contribution.
>
> - Zero-Shot Generality: Unlike methods that couple to specific sensors or require environment-specific retraining, our factorization allows deployment in unseen buildings immediately upon receiving a floor plan.
> - Interpretability: While learned BEV methods output dense feature grids, GeoReasoning produces a verifiable chain of evidence (anchors $\to$ masks $\to$ distances).
> - Human-like Map Reading: We do not aim to reinvent sensor-fusion SLAM; rather, we demonstrate that MLLMs can bridge the gap between a partial visual observation and a global map through reasoning, enabling navigation in GPS-denied, privacy-sensitive, or pre-mapping-free environments.

---

> ### Author Response · Authors · 2025-11-24
>
> > ### Q1 - Distance Estimation
>
> We agree that ultra-fine localization from a single egocentric image is intrinsically hard—even for humans (our human S@0.1 is only 0.156). Our pipeline therefore emphasizes relative/ordinal consistency across multiple anchors and solves pose with a robust trilateration objective that down-weights inconsistent ranges, much like how human localization relies on relational cues more than exact metric estimates. This design explains why, at meter-level thresholds (S@1/S@3), accuracy is driven primarily by instance disambiguation + geometry rather than tiny absolute range errors.
>
> That said, we explicitly tested the reviewer’s suggestion. In Sec 4.5 Table 2, replacing LLM range estimates with (i) ground-truth distances or (ii) a promptable monocular depth module (DepthLM) improves ultra-fine localization substantially while leaving meter-level metrics largely saturated.
>
> > ### Q2 - Failure Cases
>
> More failure cases and patterns are provided in Appx. Sec. A.1 Fig. 6.
>
> > ### Q3 - Notations
>
> We’ve made numbering consistent throughout: all displayed equations that are referenced in the text are now numbered, while the rest are definitional equalities. In addition, the reasoning segmentation (RPS) definition equation has been revised to align with the established notational conventions used earlier.

---

### Official Review · Reviewer_g4Z3 · 2025-11-03

**Soundness:** 3
**Presentation:** 3
**Contribution:** 3
**Rating:** 8
**Confidence:** 5

**Summary:**

This paper proposes a method for localization in a camera setting through its use of landmarks, done a bit differently from SLAM like approaches. They also claim to release a new dataset for indoor scenes to support the research effort. The procedure can be used as an addon to generic MLLMs and improve its performance.

The main idea is to reconcile a camera view with landmarks obtained from a 'map' - a top down view of the scene. It is a two stage process, first using intelligent prompting to get an intermediate output and then refining it further through optimization.

The map can generate anchors of interest, prompted by a multimodal LLM - a SAM like apparatus. Once they are obtained, one sets up an optimization objective through a trilateralization procedure described.

$ L = \sum w_k \varphi (||p - a_k|| - \rho_k) $

Here, we know $a_k$ from the landmarks (global solve), and $\rho_k$ is the distance in local camera. So from this we can get $p$ and thus localize. In essence we obtain p that minimizes the residual contained within.

A fairly convincing set of evaluations is provided. The gist of it is that the fittings - querying with a prompt for landmarks, and the optimization are able to significantly improve vanilla MLLM performance.

**Strengths:**

+ Intuitive way to connect indoor localization pieces
+ Camera + map makes sense - also applicable in settings like autonomous driving with BEV.
+ Principled constraint and prompting machinery.
+ Results demonstrate the correctness of the approach. We see improved results with their fittings in nearly all the cases and models. To drill down, ID switches are reduced, localization accuracy is improved.

**Weaknesses:**

- Please take this with a grain of salt. I am pretty sure that similar ideas abound in localization (for instance, in my research in BEV modelling), where we can carry out the task given a map and images by correlating them (with attention, or other means). However, the main novelty in this work is to use it with multi-modal LLMs. So to this end, I (very weakly) question the novelty of the approach.
- Temporal modelling. It would have been nice if the authors could have extended the analysis with temporal modelling. I would gather that this extension is straight forward given the methods we have today.
- More failure cases would be helpful.

**Questions:**

I am curious about what would happen if the map were erroneous. These happen in outdoor driving scenes like construction zones.
On similar lines, I am curious about scaling errors, and generally about miscalibrated maps.

---

> ### Author Response · Authors · 2025-11-24
>
> > ### W1 - Novelty
>
> We appreciate the reviewer’s perspective: correlating images with maps (often via attention in a BEV space) is well-established. Our contribution is not to re-invent correlation, but to show that training-free, test-time factorization with multimodal LLMs already yields global, floorplan-frame localization from a single egocentric image, with verifiable and explainable intermediates (anchors, masks, distances) and meter-scale metrics.
>
> Previous localization method typically (a) learns a cross-view mapping with task-specific training (and often camera calibration / multi-view sequences), (b) couples to a particular sensor/config/domain, and (c) outputs a dense grid feature map rather than an interpretable chain of evidence. In contrast, our system is zero-shot on Loc-Bench: we freeze off-the-shelf MLLMs + a segmenter, assume no odometry / depth / extrinsics, and directly predict absolute coordinates on a provided floorplan, including uncertainty regions under symmetry. This makes the approach map-agnostic (CAD scans, binary occupancy, or semantic rasters), plug-and-play across models, and interpretable—properties that are hard to obtain from a learned BEV model without re-training or domain adaptation.
>
> The zero-shot, explainable path to a global map coordinate is valuable for humanoid/robot localization in unseen buildings, rapid AR/first-responder deployment where only a floorplan is available, or privacy-sensitive settings where storing long video traces for training is undesirable. We believe our results potentially pave a clear path toward general-purpose embodied planning: zero-shot map grounding with interpretable reasoning.
>
>
> > ### W2 - Temporal Modelling
>
> We scoped this work to the single-image setting to isolate the effect of test-time factorization. As the reviewer suggests, extending to temporal input is straightforward and plug-compatible. In Appx. Sec. A.1 we add a demonstration: we sample a few frames from a short video, and run GeoReasoning per frame. Because adjacent frames share co-visible anchors, the cross-view verifier can more easily resolve instance ambiguity, and the fused estimate reduces single-frame noise—yielding smoother trajectories and improved accuracy. No additional training or calibration is required here; temporal cues are an orthogonal enhancement to our training-free pipeline.
>
>
> > ### W3 - More Failure Cases
>
> More failure cases and patterns are provided in Appx. Sec. A.1 Fig. 6.
>
>
> > ### Q1 - Error Map
>
> Our segmenter–verifier pipeline is biased toward persistent, high-salience anchors (structural cues and typically stationary large items such as built-ins, beds, sofas, tables). In daily indoor settings these anchors rarely move, and the cross-view semantic verifier relies on stable contextual cues (doorways, corridor turns, fixed cabinetry) to validate or reject a candidate region. The reasoning-segmenter and verifier introduce flexibility. When a map is outdated or locally wrong, scores for conflicting anchors drop while structure-consistent candidates remain top-ranked, so performance degrades gracefully rather than failing catastrophically.
>
> In construction-like changes (the outdoor analogue), the same principle applies: the verifier prefers permanent infrastructure (walls/door frames indoors; curbs/medians/lane geometry outdoors), treats temporary artifacts (cones, signage, moved furniture) as low-trust cues, and—in case of broad inconsistency—we need to update the map.

---

### Official Review · Reviewer_kFu9 · 2025-11-06

**Soundness:** 1
**Presentation:** 2
**Contribution:** 2
**Rating:** 2
**Confidence:** 3

**Summary:**

This paper introduces GeoReasoning, a new zero-shot framework for self-localization that requires no sensors or environment-specific training. It uses multimodal large language models (MLLMs) to interpret maps and visual observations. The method works by first identifying landmarks with an MLLM, locating them in an image, estimating their distance through textual reasoning, and then using trilateration to find the user's position. The paper also introduces Loc-Bench, a new evaluation dataset, on which GeoReasoning outperformed direct MLLMs prompting methods.

**Strengths:**

The framework is interpretable and modular, separating semantic reasoning from geometric computation. Using semantic anchors followed by trilateration mirrors human reasoning and makes the multimodal localization process more transparent and analyzable.

**Weaknesses:**

Main Weakness:
1. The author states that they create Loc-Bench that differs from previous work by focusing on egocentric-to-allocentric transformation, while VSI-Bench[1] have included a 2D cognitive map and evaluates visual-spatial relationship from egocentric videos, espatially testing on relational reasoning
and egocentric-allocentric transformation.
2. The baselines used for comparison are limited. There are more SOTA open-sourced VLM that could be compared to, e.g. InternVL2.5 and LLaVA, which would better contextualize the improvements. I also suggest the authors to provide baselines for (1) always selecting the most frequent answer; (2) a random selection strategy, which would help quantify the task’s inherent difficulty. Besides, a human-level performance would establish an intuitive upper bound and clarify the practical significance of the results.
3. The ablation study does not evaluate each component's strength. Key design choices (e.g. anchor reasoning, verification and refinement) are not individually quantified, leaving it unclear which parts actually drive performance. A more fine-grained ablation would clarify whether the observed improvements arise from the overall pipeline design or a few dominant modules.



Minor:
1. Only one equation is indexed; all others are not. Please ensure consistent numbering.
2. In the equation on line 250, what does $\mathcal{D}$ represent? Is it a typo for $G$? Also, the use $\mathbb{R} > 0$ is not formal.
3. Some notations are undefined, e.g., the $L$ in Equation (1). Define all symbols on first use.


[1]Yang, Jihan, et al. "Thinking in space: How multimodal large language models see, remember, and recall spaces." Proceedings of the Computer Vision and Pattern Recognition Conference. 2025.

**Questions:**

In lines 220–238, how does the cross-view semantic verifier handle repeated object categories (e.g., multiple bathrooms, beds, or tables), given that prompt-based segmentation on $M$ may highlight only one instance among several plausible candidates?

**Details Of Ethics Concerns:**

None.

---

> ### Author Response · Authors · 2025-11-24
>
> > ### W1 - VSI Bench
>
> We appreciate R1’s pointer to VSI-Bench. As mentioned in Section 2.2, VSI-Bench is a video-based VQA benchmark across configuration / measurement / spatiotemporal tasks. It further prompts models to draw “cognitive maps” and evaluates them on a coarse grid (e.g., 10×10/20×20) to study local vs. global spatial memory. As the authors report, explicit cognitive-map prompting helps distance QA, but the maps that MLLMs produce tend to be locally accurate yet globally weak—reflecting the partial coverage inherent to a single video trajectory.
>
> By design, Loc-Bench targets a different tasks: given a single egocentric RGB and a  floorplan, the system must return a 2D allocentric coordinate (or region) in the map frame. We score coordinate-level performance with VPR, meter-scale localization error, S@r (0.1/0.5/1/3 m), signals unavailable in VSI-Bench’s QA setting. This makes allocentric grounding quantitative in meters, rather than indirectly inferred via QA accuracy or low-resolution grids.
>
> VSI-Bench primarily probes local/short-range spatial relations along an egocentric video trajectory (and on low-resolution grids), i.e., whether a model can retain and relate nearby layout facts. Loc-Bench, in contrast, evaluates global allocentric reasoning: from a single view, the model must infer its absolute position on the entire floorplan, respect navigability, and achieve meter-level accuracy. Both capabilities are essential for a full assessment of spatial intelligence—VSI-Bench covers the local relational QA side, while Loc-Bench provides the complementary, coordinate-level evaluation of global map reasoning. Our benchmark is an initial or indeed pioneering exploration of this global aspect in a training-free setting.
>
>
> > ### W2 - More VLMs & Human
>
> We thank the reviewer for the suggestions and we include more VLMs to compare with and human performance as an upper bound. Please find the experiment results in Sec 4.3 Table 1.
>
>
> > ### W3 - Ablation study
>
> We thank the reviewer for the suggestions and we conduct comprehensive ablation study on each sub-module. Please find the experiment results in Sec 4.5 Table 2. To summarize their individual contributions:
> - Cross-view verification is the single largest driver of accuracy at 0.5-3m, preventing instance swaps and relational violations in symmetric layouts.
> - Refinement stabilizes masks and helps at tight thresholds, with smaller impact at 1-3m.
> - Better distances (DepthLM or GT) principally improve S@0.1 and slightly reduce median error, indicating that perception noise manifests first at ultra-fine scales.
> - Factorization itself  (vs. direct prompting) yields order-of-magnitude gains, demonstrating that structured reasoning + geometry is the core lever—independent of training.

---

> ### Author Response · Authors · 2025-11-24
>
> > ### Q - Verifier
>
> In practice, our reasoning-prompted segmentation enumerates all instances that satisfy the prompt, not just one; i.e., for a prompt like “bathroom sink,” the segmenter typically returns multiple mask proposals covering each plausible instance on the floorplan (see Fig. 3-left). The cross-view semantic verifier is then applied to each candidate mask to resolve which instance corresponds to the egocentric observation. Concretely, for a map-crop around a candidate, the verifier checks contextual consistency between the first-person view and the map neighborhood, including: (i) co-visible object cues (objects seen in FPV appear near the candidate on the map), (ii) relational and directional constraints (e.g., “chair to the right of table,” relative orientation to doors/corridors), and (iii) layout plausibility (room boundaries and type). Candidates that fail these cross-view checks receive low scores and are discarded; the highest-scoring candidate is selected as the grounded instance.
>
> For symmetric layouts with near-identical rooms (e.g., multiple bathrooms), the verifier can be uncertain if local context is indistinguishable. Our multi-rotation scheme (R=3) mitigates this by rotating the egocentric reference and re-verifying with a larger contextual ring, allowing the verifier to exploit out-of-room references (e.g., a nearby corridor turn or a distinctive object outside the bathroom) to break symmetry. This reduces—but does not eliminate—ambiguities.
>
> As shown in Appx. Fig. 6, in extremely symmetric cases (e.g., identical hotel rooms repeated along a corridor), even human annotators cannot uniquely localize the view. These locations are isometrically indistinguishable under our experiment settings. These are therefore information-limited rather than verifier-limited failures.
>
>
> > ### Minor
>
> 1. Equation numbering. We’ve made numbering consistent throughout: all displayed equations that are referenced in the text are now numbered, while the rest are definitional equalities.
>
> 2. $\mathbb{D}$ denotes our LLM-based distance estimator. We clearly defined in the revised version. And we use $\mathbb{R}_{>0}$ to denote real positive numbers.
>
> 3. We now define all symbols on first use. In particular, $L$ is the loss in our robust trilateration objective.

---

### Author Response · Authors · 2025-11-24
**Rebuttal to all reviewers**

We thank our ICLR reviewers for their thoughtful feedback, especially g4Z3 for understanding and appreciating **GeoReasoning** in high confidence.
Our work targets a precise and practical setting: single egocentric RGB $\rightarrow$ global allocentric coordinate on a provided floorplan, with no task-specific finetuning and verifiable intermediates (anchors, masks, and ranges).

Georeasoning is one of the first to do "map  + camera" (g4Z3), and so we pledge the other reviewers *don’t let the perfect be the enemy of the good*, to quote an old saying, causing the ICLR community to miss out on making progress with a good enough solution that will likely spark future worthwhile research on **GeoReasoning**.

These imperfections, rightly pointed out by our reviewers, including benchmarks and baseline comparison (kFu9), dynamic objects (PKvn),
employment rather than mitigating existing retrieval-based tools and comparison on different LLMs (YuzH), are addressed in the following and the revised paper as well.

An enhanced version of the paper revised to address reviewers' questions has been uploaded, with modified sections marked in blue. Below is a summary of changes:
- A comprehensive ablation with discussion (Sec 4.5)
- A proof-of-concept example demonstrating that GeoReasoning can be extended to outdoor scenes (Sec 4.6).
- Demos showing capabilities in temporal modeling
(Appx. Sec A.1).
- More failure case analysis (Appx. Sec A.1).
- Minor changes in paper writing.

### More Ablations
We conducted additional ablations as suggested by the reviewers. The relevant experimental results are presented in our {\em revised paper}, located in Sec 4.5 Table, which validates the effectiveness of individual components of GeoResoning. To summarize their respective, individual contributions:
- Cross-view verification is the single largest driver of accuracy at 0.5–3m, preventing instance swaps and relational violations in symmetric layouts.
- Refinement stabilizes masks and helps at tight thresholds, with smaller impact at 1–3m.
- Better distances (DepthLM or GT) principally improve S@0.1 and slightly reduce median error, indicating that perception noise manifests first at ultra-fine scales.
- Factorization itself  (vs. direct prompting) yields order-of-magnitude gains, demonstrating that structured reasoning + geometry is the core lever—independent of training.

### Outdoor Scenes
Although Loc-Bench is indoor, **GeoReasoning** transfers zero-shot to outdoor scenes. Using a satellite BEV as the "floorplan" and a single egocentric frame, our pipeline selects permanent landmarks, estimates their ranges, and localizes via robust trilateration—landing in the correct region with only a minor prompt tweak. This proof-of-concept supports the method’s map-framed, domain-agnostic nature; we plan to release an outdoor split with diverse landmarks.

---

### Author Response · Authors · 2025-12-01
**Global Response**

We are grateful that Reviewer g4Z3 (rating 8, confidence 5) explicitly endorses the work as a good paper, highlighting its principled design, intuitive “camera + map” formulation, and consistent improvements over vanilla MLLM prompting. Their assessment is based on strong familiarity with BEV/localization literature and a careful check of the technical details, and they view MLLM-based reasoning localization as a promising direction beyond traditional SLAM-style pipelines. We respectfully ask the AC to weigh this high-confidence champion’s view alongside the more skeptical reviews, which focus mainly on scope and comparative framing rather than on correctness or lack of contribution.

During rebuttal we substantially strengthened the paper to address these concerns. Concretely, we (1) added fine-grained ablations (Sec. 4.5) that quantify the contribution of each component (factorization, cross-view verification, refinement, and improved distances), (2) included additional VLM baselines and human performance to better contextualize our metrics, (3) provided proof-of-concept outdoor examples (Sec. 4.6) using satellite BEV maps to demonstrate that the method is not restricted to indoor layouts, and (4) expanded temporal modeling demos and failure-case analysis in the appendix, clarifying how temporal cues can be plugged in without retraining and how the system behaves under map errors and symmetries. These additions directly respond to the main weaknesses raised and, we believe, resolve the most serious doubts about novelty, robustness, and generality.

Our submission GeoReasoning: Structured Semantic Reasoning for Image-to-Map Localization introduces (i) a new reasoning-localization paradigm that factorizes image-to-map grounding into structured semantic reasoning plus robust trilateration, and (ii) Loc-Bench, the first benchmark that evaluates single-image egocentric-to-floorplan localization in meter-level coordinates with verifiable intermediate reasoning (anchors, masks, ranges). This zero-shot, training-free pipeline shows that off-the-shelf MLLMs can already perform global map localization from a single RGB frame with interpretable evidence, which we believe is a timely and distinctive contribution for ICLR’s embodied AI and reasoning communities.

All reviewers acknowledged that our two-stage “Reason & Ground / Constrain & Solve” framework is sound and that the new benchmark is meaningful. Our key conceptual uniqueness is how we use geometry: rather than retrieval over pre-built galleries or environment-specific BEV models, GeoReasoning generates and tests map-space hypotheses at test time via MLLM-driven landmark selection and cross-view verification, then solves pose with robust trilateration—fully zero-shot, with no environment-specific training, odometry, or 3D reconstruction. This yields a practical, human-like pipeline for unseen buildings and privacy-sensitive or rapidly changing environments where building and maintaining retrieval bases is impractical.

In summary, GeoReasoning offers a distinct, training-free, interpretable route to global map localization with MLLMs and introduces a broad, reproducible benchmark for this emerging task. It opens a concrete, experimentally validated path for future work on spatial reasoning, map grounding, and embodied planning with large multimodal models.

---

### Meta-Review · Area_Chair_7sXq · 2026-01-04

**Summary:**

This paper presents an approach for using a MLLM to perform image-to-map localization via a series of structured reasoning queries that are performed in a zero-shot framework.  Some key concerns of the reviewers include a question of how different the proposed task is as well as the benefits over prior work.  These related to the core contributions of the paper, and three of the four reviewers gave initial recommendations for rejection.  As I will expand on further below, key issues remain unresolved and should be corrected before this paper is published.

**Reviewer Concerns:**

Some concerns related to ablations or performance of other MLLMs were addressed adequately in the rebuttal However, some concerns I outlined in the summary were mostly argued without experiments, despite explicit requests.  Namely, the reviewers asked for direct comparisons to prior work.  The arguments provided by the authors does little to assuage the concerns as presented.  E.g., the authors claim that using a limited FOV image rather than a panoramic image would reduce performance.  This may very well be the case, but unless this has been experimentally validated it is just conjecture.  While I do appreciate the sentiment raised by the authors that adapting prior work may be time consuming, the overwhelming feedback they are receiving is that at least these sets of reviewers find it necessary to demonstrate a contribution.  Further, while the authors claim that prior work may not have realistic assumptions, this can be easily countered with an argument saying that assuming you can use a MLLM in robotics is not realistic as well- they take too much compute.  This is true even if we were to use a basic query even once let alone to perform the complicated reasoning steps the authors proposed in their method.  While better compute may eventually solve this issue, I mostly highlight this to illustrate the shortcomings of the authors arguments.

Some other statements made by the authors were not well supported.  For example, the a reviewer raised an issue that the authors did not provide evidence to support that their approach can work in outdoor environments.  The authors then pointed to an example in their paper, but what they showed nothing more than anecdotal rather than proper evidence.

**Reviewer Scores:**

I believe the reviewers generally may have increased their scores as several of the stated issues have been resolved.  However, the authors do not compare to prior work, and their discussion on the subject is not convincing.  As such, I find it unlikely that a majority of reviewers would have raised their scores to recommend acceptance.

---

### Decision · Program_Chairs · 2026-01-26

Reject